# Strongyloidiasis in Southern Alicante (Spain): Comparative Retrospective Study of Autochthonous and Imported Cases

**DOI:** 10.3390/pathogens9080601

**Published:** 2020-07-23

**Authors:** Ana Lucas Dato, María Isabel Pacheco-Tenza, Emilio Borrajo Brunete, Belén Martínez López, María García López, Inmaculada González Cuello, Joan Gregori Colomé, María Navarro Cots, José María Saugar, Elisa García-Vazquez, José Antonio Ruiz-Maciá, Jara Llenas-García

**Affiliations:** 1Internal Medicine Department, Hospital Vega Baja, 03314 Orihuela, Spain; belenmarlo2005@hotmail.com (B.M.L.); mariagarc.lopez@gmail.com (M.G.L.); gonzalez_inmcue@gva.es (I.G.C.); gregori_juacol@gva.es (J.G.C.); 2Foundation for the Promotion of Health and Biomedical Research of Valencia Region (FISABIO), 46020 Valencia, Spain; borrajo_emi@gva.es (E.B.B.); navarro_dia@gva.es (M.N.C.); 3Internal Medicine Department, Hospital Clínico Universitario Virgen de la Arrixaca, 30120 Murcia, Spain; misabel.pacheco@carm.es; 4Microbiology Department, Hospital Vega Baja, 03314 Orihuela, Spain; 5Parasitology Department, Centro Nacional de Microbiología, Instituto Carlos III, 28903 Madrid, Spain; jmsaugar@iscii.es; 6Infectious Diseases Unit, Hospital Clínico Universitario Virgen de la Arrixaca, 30120 Murcia, Spain; elisagv@um.es; 7Instituto Murciano de Investigación Biosanitaria (IMIB), Universidad de Murcia, 30120 Murcia, Spain; 8Pathological Department, Hospital Clínico Universitario Virgen de la Arrixaca, 30120 Murcia, Spain; josea.ruiz14@carm.es; 9Clinical Medicine Department, Universidad Miguel Hernández de Elche, 03202 Elche, Spain

**Keywords:** *Strongyloides stercoralis*, strongyloidiasis, immigrants, immunosuppression, eosinophilia, ivermectin

## Abstract

Background: Strongyloidiasis is a parasitic disease with global prevalence. In Spain, autochthonous cases are concentrated in the Mediterranean basin. We aimed to analyze clinical and epidemiological characteristics of *Strongyloides stercoralis* infection in Vega Baja del Segura (Spain), comparing autochthonous versus imported cases. Methods: Observational retrospective study of all strongyloidiasis cases from January 2009 to January 2019. Cases were diagnosed by stool larvae visualization, positive culture, PCR, *Strongyloides* serology, and/or compatible histology. Results: We included 36 patients (21 men) with a mean age of 60.8 years ±17.6; 15 cases were autochthonous and 21 imported 80.9% from Latin America. Autochthonous cases were associated with older age (mean 71.3 vs. 53.3 years; *p* = 0.002), male sex (odds ratio (OR) 5.33; 95% confidence interval (CI) 1.15–24.68; *p* = 0.041), and agricultural activity (OR 13.5; 95% CI 2.4–73.7; *p* = 0.002). Fourteen were asymptomatic, three autochthonous cases presented with hyperinfection syndrome, and two patients died. There was no difference between autochthonous versus imported origin in eosinophilia at diagnosis (93.3% vs. 75%; *p* = 0.207), treatment received, or clinical response (85.7% vs. 88.9% cured; *p* = 1). Conclusion: In our region, imported strongyloidiasis coexists with autochthonous cases, which are mainly in older male farmers who are diagnosed at more advanced stages. Systematic screening programs are needed.

## 1. Introduction

*Strongyloides stercoralis* is an intestinal helminth acquired by humans when larvae penetrate intact skin following contact with infected soil [1]. It has a peculiar autoinfection life cycle that may lead to lifelong chronic infection if left untreated. Its clinical spectrum can range from cutaneous (pruritus, larva currens, urticaria), digestive (diarrhea, abdominal pain), or respiratory (chronic cough, dyspnea) symptoms to asymptomatic eosinophilia or even silent infections, which constitute the vast majority of infections in endemic areas [2,3].

On occasion, it manifests as a life-threatening complication with the development of hyperinfection syndrome. While initially confined to the lungs and the gastrointestinal tract, it has signs and symptoms of severe disease due to an elevated number of larvae. With time, this syndrome can progress toward a disseminated strongyloidiasis, which is the most severe form, when larvae might be found in any organ, not just in the respiratory or gastrointestinal tract. Both the hyperinfection and disseminated forms are particularly common in immunosuppressed patients, such as those with hematological malignancies, those with human T-lymphotropic virus type 1 (HTLV-1) infection, transplant recipients, and those receiving immunosuppressive treatments (especially corticosteroids) [1,4]. 

Strongyloidiasis is prevalent globally, with an estimated 30 million to 370 million infected people worldwide [5]. Although it generally occurs in tropical and subtropical regions, sporadic cases have been reported in temperate countries [6]. Despite its high prevalence, it has not been well studied. Furthermore, its prevalence may be underestimated due to the lack of a gold standard diagnostic test [4,7].

In Spain, increased migratory flows have driven recent increases in imported cases [8]. Prevalence amongst immigrants is variable, ranging from 4.5% to 17% in studies done in referral units but up to 36% in people coming from Cambodia or 10% to 27% in those from Latin America [9,10].

Autochthonous cases also occur in Spain, especially in the Mediterranean basin (Valencian region and Murcia). The vast majority are restricted to a specific area of La Safor and Marina Alta regions, corresponding to Health Area 11 of the Valencian Community [11,12]. In these areas, prevalence is 0.3% in the general population and 12.4% in high-risk groups, mostly men aged over 50 who participate in agricultural activities [13]. A combination of adequate temperature and humidity, lack of hygiene in rural areas during the 1960s, and a population exposed to *S*. *stercoralis* for occupational reasons (such as rice farmers or irrigation ditch cleaners) may have facilitated a concentration of cases in that particular area. A greater awareness of strongyloidiasis among health care workers of that department may have contributed as well [14]. Outside this region, only a few isolated autochthonous cases have been reported, most of which have been based only on serological diagnosis [14,15,16,17]. In light of these limited case numbers, Spain is not considered an endemic country for strongyloidiasis [18]. 

Vega Baja del Segura is a region located in southern Alicante, near the border with Murcia, where some isolated cases have been reported [19]. In a previous study, our group retrospectively reviewed the cases of autochthonous strongyloidiasis at the Vega Baja Hospital in Orihuela between January 1999 and March 2016, finding 10 autochthonous cases in that period [20]. 

Our region thus offers unique opportunities for strongyloidiasis research, allowing comparisons between imported and autochthonous cases. The aim of this study was to analyze the clinical, epidemiological, and microbiological characteristics of all patients diagnosed with strongyloidiasis at the Vega Baja Hospital between 1999 and 2019 and to compare autochthonous versus imported cases. 

## 2. Results

Between January 1999 and January 2019, 36 patients (21 men, 58%), with a mean age of 60.81 ± 17.57 years, were diagnosed with strongyloidiasis at the Vega Baja Hospital. There were 15 autochthonous and 21 imported cases (Figure 1).

Eighteen of the imported cases came from Latin America (1 Dominican Republic, 1 Ecuador, 1 Colombia, 12 Bolivia, 1 Jamaica, and 1 English woman who frequently travelled to the Caribbean) and 4 from Africa (3 Morocco and 1 Spanish long-term resident in Algeria). The geographic distribution of autochthonous and imported patients according to the municipality of residence in the Vega Baja del Segura region is shown in Figure 2.

Twenty-two cases (61.1%) were symptomatic at the time of diagnosis, most of them (n = 9) with mild digestive symptoms such as diarrhea and abdominal pain, followed by pruritus (n = 3) and respiratory symptoms (n = 2). Two autochthonous cases had a hyperinfection syndrome, and one had disseminated strongyloidiasis; all had been treated with corticosteroids prescribed for different comorbidities (two for chronic obstructive pulmonary disease and one for lung cancer); two died.

Six patients (16.7%) had a neoplasia (two bladder tumors, one lung adenocarcinoma, one colon adenocarcinoma, one meningioma, and one non-Hodgkin’s lymphoma), and three patients (8.3%) had chronic obstructive pulmonary disease. Ten patients (27.8%) were co-infected with *Trypanosoma cruzi* and one patient with hydatidosis. Twenty-nine patients were serologically tested for HIV, all with negative results, and only five (two autochthonous and three imported) were tested for human T cell lymphotropic virus 1 (HTLV-1), also with negative results. Five patients were treated with corticosteroids at the time of diagnosis (26.7% autochthonous vs. 4.8% imported). Table 1 shows the comparative analysis between autochthonous and imported patients.

Eosinophilia at diagnosis was detected in 29 patients (80.6%): in 14 (93.3%) of the autochthonous cases and 16 (75%) of the imported ones, with an average eosinophil count of 1200 ± 870 eosinophils/mm^3^. *Strongyloides* serology was performed in 34 patients (94.4%) and was positive in all of them; 22 cases (64.7%) had a serological index greater than 2.5 (12 imported and 10 autochthonous). *Strongyloides* stool culture was performed in 20 patients (55.5%) and was positive in three. In four patients, the diagnosis was made by visualization of larvae in a biopsy, in one by duodenal aspirate culture, and in another one by sputum culture.

Twenty-seven patients were treated with ivermectin (24 with double dose two weeks apart, 2 with a 7-day course, and 1 with a 14-day course), three with ivermectin plus albendazole, one with albendazole, and one with thiabendazole. Four (11%) patients died, and three (8%) had a therapeutic failure and were treated again (two autochthonous and one imported case). The two autochthonous cases with therapeutic failure were immunosuppressed patients: one was treated initially with ivermectin double dose and after failure with ivermectin plus albendazole for seven days, and the other one with ivermectin double dose in both occasions. The imported case with therapeutic failure was not immunosuppressed and was treated on both occasions with ivermectin double dose. Table 2 and Table 3 summarize the clinicoepidemiological characteristics of autochthonous and imported cases. 

## 3. Discussion

In Spain, autochthonous cases of strongyloidiasis are concentrated in the Mediterranean basin, namely in the autonomous regions of Valencia and Murcia, where autochthonous cases coexist with imported strongyloidiasis. In our region, autochthonous cases are predominantly in men and appear in people who are older compared to those with imported cases. Agricultural activities, whether for work or for leisure, were also associated with autochthonous cases. The sociodemographic characteristics of our autochthonous patients are similar to those described in previous studies in the Valencian region [13,20]. Moreover, a recent systematic review of autochthonous cases [21] of strongyloidiasis in Spain also reports a high percentage of men (82.9%), agricultural occupations, and a mean age of 68.3 years. The predominance of men among the autochthonous cases seems to be linked to the mode of acquisition, in most cases occupational. On the other hand, there are some studies in experimental animal models [22] that describe a greater susceptibility to *Strongyloides* in male rats, attributed to the protective effect of estrogen and the immunosuppressive role of androgens. Autochthonous patients had a mean age of 71.3 ± 16.2 years; if screening of the autochthonous population is considered, the target population should include at least people aged over 55 years.

In our study we found a higher percentage of symptomatic patients among autochthonous cases, although this result was not statistically significant. This may be due to a lower rate of suspicion in autochthonous patients, who are often diagnosed after presenting symptoms, while immigrants are more frequently screened opportunistically in clinical settings or in community screening campaigns for Chagas disease [23]. Likewise, the only three cases of hyperinfection syndrome occurred in autochthonous cases. Patients with autochthonous strongyloidiasis also had more comorbidities and more risk factors for hyperinfection (especially corticosteroid treatment).

In our study, as in others [12,24,25], the prevalence of eosinophilia was high in both groups, representing a good indicator of infection. However, restricting the screening to people with eosinophilia could result in missed cases, especially in patients already on corticosteroid treatment in whom eosinophilia may be more frequently absent.

Regarding strongyloidiasis diagnosis, the main problem is the low sensitivity of traditional diagnostic methods and the absence of a gold standard. The sensitivity of techniques for detecting rhabditiform larvae in feces does not exceed 40%, due primarily to the intermittent pattern of excretion [5]. Agar plate culture of *Strongyloides* larvae increases the sensitivity to some extent, but it remains under 60% in single samples. Serology (IgG for filariform larvae) has a better sensitivity (83–89%) and is considered the best technique to screen asymptomatic immigrants and monitor treatment effectiveness. A recent study on the diagnostic accuracy of serological tests for *S. stercoralis* reported high sensitivity (91.2%) and specificity (99.1%) for IVD-ELISA [26]. However, another study in immunosuppressed patients showed a sensitivity of just 42.9%, with a specificity of 96.3% [27]. The latest guidelines for the management of strongyloidiasis in non-endemic countries recommend a combination of serological and parasitological techniques in immunosuppressed patients [28]. Serology can yield false positives, mainly by cross-reaction with other helminths; however, a recent study shows that increasing the optical density index (OD)index cutoff to 2.5 can increase specificity to almost 100% [26]. Applying this cutoff to our series would result in 26 cases: 22 cases diagnosed by serology (12 imported and 10 autochthonous), plus 4 cases with parasitological diagnosis and no serology available. The use of *Strongyloides* RT-PCR in feces seemed promising in early studies, with sensitivity above 90% [29], but it has not proven to add much value to serology. A recent meta-analysis reports a PCR sensitivity of 56.5% to 71.8% and concludes that further studies are needed to determine its real diagnostic utility [4].

Regarding treatment, in our series most cases were treated with ivermectin, with high efficacy rates and excellent tolerability. No statistically significant differences were observed between autochthonous versus imported cases regarding treatment or treatment response (response 85.7% vs. 88.9%; *p* = 1). These cure rates are very similar to those described in the Strong–Treat study, which compared the efficacy of a single dose of ivermectin versus a four-dose regimen, reporting an efficacy of 86% for the single-dose regimen [30]. Three patients required retreatment for therapeutic failure, two autochthonous and one imported.

In the municipality of Oliva (Valencia province), one study estimated seroprevalence at 0.9%, with authors concluding that there is probably no active transmission at present given the absence of cases in young people and the changes in agricultural practices [13]. The systematic review of endemic cases in Spain [22] showed a decrease in the number of cases reported since 2011. However, in our series, most cases were diagnosed in recent years. This seems to be due to a higher rate of suspicion among professionals in the area following several educational activities about the existence of autochthonous cases in the health department. Although our patients’ characteristics are similar to those from Oliva, we cannot categorically state that there is no active transmission in our area. In our opinion, large seroprevalence studies and field studies searching for larvae would be needed to rule out active transmission.

Given the potential severity of strongyloidiasis in immunosuppressed individuals, we believe it is essential to implement a strongyloidiasis screening protocol in our area, both in immigrants from and travelers to endemic areas [28], along with autochthonous people with risk factors, especially those who are becoming immunosuppressed. A recent cost-effectiveness study of strongyloidiasis screening in immigrant populations [31] in Europe concluded that prophylactic administration of ivermectin is cost-saving in patients from strongyloidiasis-endemic countries and at risk of immunosuppression. However, additional studies are needed on whether different screening strategies in autochthonous populations could be more cost-effective than prophylactic treatment.

The present study has several limitations. First, since our setting has sporadic cases of strongyloidiasis, we cannot rule out that some cases in long-term residents of the Vega Baja del Segura area were misclassified as imported. Another limitation of our study is the use of different diagnostic methods over time and the fact that in a significant percentage of cases, diagnosis was exclusively serological. Besides, Latin American countries are endemic for HIV and HTLV-1 virus. HTLV-1 infection is a well-known factor for strongyloidiasis hyperinfection syndrome [32]. However, information about HTLV-1 serostatus of our patients was limited as serology was not routinely requested. Furthermore, the study’s retrospective nature means that case follow-up was not standardized, and treatment response has been assessed at different times. Serology is a frequently used response marker that has proven useful in several studies, including in immunosuppressed individuals [33,34].

## 4. Materials and Methods 

Observational retrospective study in all patients diagnosed with strongyloidiasis at the Vega Baja Hospital (Orihuela) from January 2009 to January 2019 was used. The Orihuela health department has a population of 167,415 inhabitants (33.43% foreigners) [35]. We included all cases with histopathological diagnosis (visualization of *Strongyloides* larvae in intestinal biopsy; Figure 3), parasitological diagnosis (either by visualization of larvae in fresh stool or after fecal culture, of duodenal aspirate, or other samples), or serological diagnosis (positive for IgG antibodies to *Strongyloides*). The serological study was carried out at the Spanish National Centre for Microbiology, Instituto Nacional de Salud Carlos III Majadahonda, Madrid. The serological study was carried out at the National Centre of Microbiology using a non-automated IVD-ELISA technique that detects IgG antibodies against raw antigen of filariform larvae (DRG Instruments Gmbh, Marburg, Germany), with a positive optical density index (OD) cutoff of 1.1. Coproparasitological and fecal culture were performed at the microbiology laboratory of the Vega Baja Hospital. Stool concentration was performed using the Mini Parasep SF Alcorfix system. *Strongyloides* culture was seeded in Mueller Hinton agar, incubated at 28 °C, and observed for five days in search of the sinuous trajectories. We considered cases autochthonous if they were in patients who had resided in the Vega Baja region for more than 30 years and had never traveled to a *Strongyloides*-endemic area, while cases were defined as imported if they were in immigrants from or travelers to endemic areas. Chronic corticosteroid treatment was defined as at least 20mg per day of prednisone for ≥14 days.

During the study period, three Chagas disease and *Strongyloides* community screening campaigns among Latin American immigrants were performed at our province, in 2016, 2017, and 2018.

Demographic, clinical, analytical, microbiological, and treatment-related variables were recorded for included patients, and their most recent status (up to 31 January 2019) was reviewed. Eosinophilia was defined as an eosinophil count exceeding 500 eosinophils/mm^3^; serological response was defined as seroreversion or post-treatment/pre-treatment OD of less than 0.6.

The analysis was performed using SPSS v. 23.0 (SPSS statistics for Windows, version 23.0, Armonk, NY, USA). The one-sample Kolmogorov Smirnov test was used to assess the distribution of continuous variables, and the Student t-test to compare them between groups. The chi-squared test or Fisher’s exact test were used, as appropriate, to compare qualitative variables.

## 5. Conclusions

Autochthonous and imported cases of strongyloidiasis coexist in our region, and diagnosis of the disease has increased in recent years. Autochthonous cases occur mainly in older men who take part in agricultural activities, and they tend to be diagnosed in more advanced stages. Increased clinical suspicion of this entity is warranted, as are screening programs for both autochthonous populations and immigrants from endemic areas. Screening could decrease the diagnostic delay and thus avoid the potential development of hyperinfection syndrome.

## Figures and Tables

**Figure 1 pathogens-09-00601-f001:**
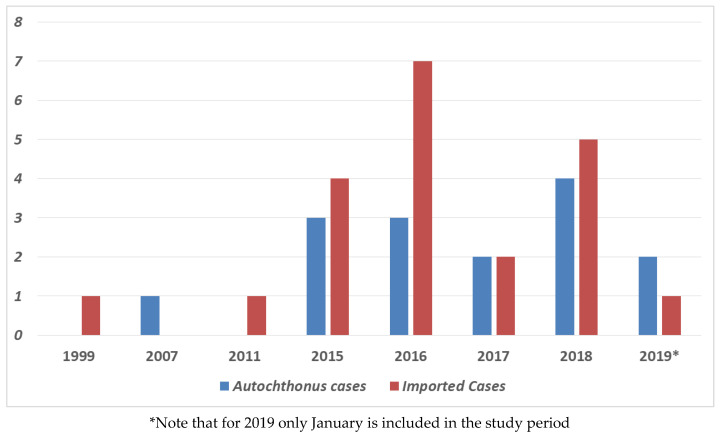
Number of autochthonous and imported strongyloidiasis cases per year in Vega Baja del Segura.

**Figure 2 pathogens-09-00601-f002:**
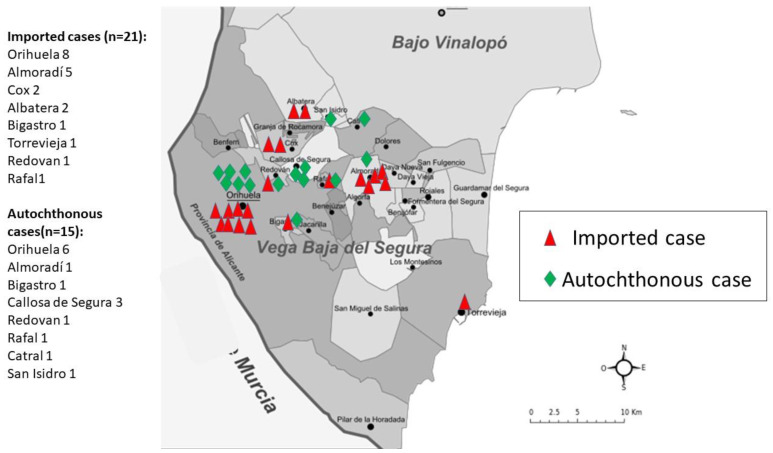
Distribution of cases of autochthonous and imported strongyloidiasis by place of residence in the Vega Baja del Segura region.

**Figure 3 pathogens-09-00601-f003:**
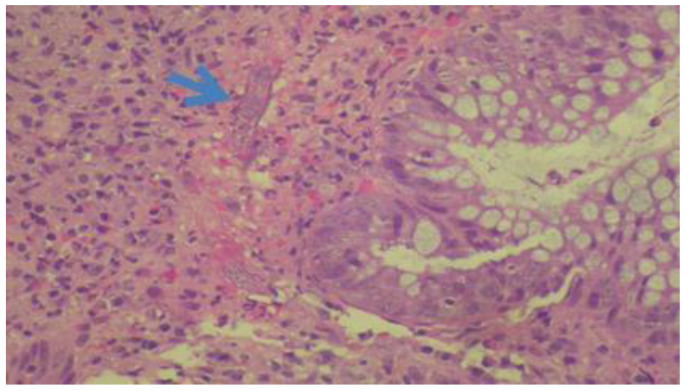
Intravascular *Strongyloides* larva (arrow) in a colon biopsy.

**Table 1 pathogens-09-00601-t001:** Comparative analysis between autochthonous and imported strongyloidiasis cases at Vega Baja Hospital.

Variables	Autochthonous (N = 15)	Imported (N = 21)	*p*	OR (95% CI)
Median age, in years	71.33	53.29	*p* = 0.002	
Men	12 (80%)	9 (43%)	*p* = 0.041	5.33 (1.15–24.68)
Agriculture dedication	10 (69.2%)	3 (14.3%)	*p* = 0.002	13.5 (2.4–73.7)
Symptomatic patients	11 (73.3%)	11 (52.4%)	*p* = 0.302	0.4 (0.75–2.12)
Corticosteroid treatment	4 (26.7%)	1 (4.8%)	*p* = 0.138	5.77 (0.55–60.60)
Eosinophilia	14 (93.3%)	16 (75.5%)	*p* = 0.207	4.66 (0.48–45.04)
Response to treatment	13 (85.7%)	19 (88.9%)	*p* = 1	3.20 (0.22–45.19)
Hyperinfection syndrome	3 (20%)	0 (0%)	*p* = 0.064	0.80 (0.61–1.03)
Death attributed to strongyloidiasis	2 (13.3%)	0 (0%)	*p* = 0.167	0.86 (0.71–1.05)

OR: odds ratio; CI: confidence interval.

**Table 2 pathogens-09-00601-t002:** Clinicoepidemiological characteristics of autochthonous strongyloidiasis cases at Vega Baja Hospital.

Case	Sex, Age in Years	Date of Diagnosis	Occupation	Comorbidities and Risk Factors	Clinical Manifestations	Eosinophilia (Eosinophils/mm^3^)	Method of Diagnosis (in Serology, Optical Density Index)	Strongyloidiasis Treatment	Outcome
1	M, 69	July 2007	Farmer	Lung cancer Chemotherapy and radiotherapy	Hemoptysis, Hyperinfection syndrome	Yes (600)	Larvae in sputum Serology not requested	None	Death
2	M, 72	March 2015	Shipper	COPD, Corticosteroids	Hyperinfection syndrome	Yes (2400)	Larvae in large intestine biopsy and fecal samples Serology +(8.88)	Ivermectin + albendazole	Death
3	F, 73	May 2015	Farmer	Diverticulosis	Digestive	Yes (2700)	Serology +(1.93)	Ivermectin + albendazole	Recovery
4	M, 80	December 2015	Farmer	Bladder cancer	Digestive, Skin	Yes (700)	Serology +(4.45)	Ivermectin	Seroreversion
5	M, 85	May 2016	Farmer	Biliary disease Low-grade lymphoma	Digestive	Yes (570)	Serology +(6.83)	Ivermectin	Re-treatment (ivermectin)
6	M, 79	June 2016	Unknown	None	Digestive	No	Serology +(3.15)	Ivermectin	Re-treatment (ivermectin +albendazole)
7	M, 71	November 2016	Gardener	Colon adenocarcinoma	Asymptomatic	Yes (2400)	Larvae in large intestine biopsy and fecal samples Serology +(8.88)	Ivermectin	Seroreversion
8	M, 80	March 2017	Farmer	COPD, Atrial fibrillation Diabetes	Disseminated Strongyloidiasis Meningitis	Yes (640)	Larvae in fecal cultures and fresh fecal samples. Serology +(1.16)	Ivermectin	Seroreversion
9	M, 83	May 2017	Farmer	Prolactinoma, Diabetes	Asymptomatic	Yes (1450)	Serology +(6.85)	Ivermectin	Dead (surgery)
10	F, 49	January 2018	Unknown	Rheumatoid arthritis	Asymptomatic	Yes (780)	Serology +(1.35)	None	Lost to follow-up
11	M, 74	May 2018	Farmer	Renal insufficiency	Skin	Yes (900)	Serology +(3.41)	Ivermectin	Seroreversion
12	F, 25	June 2018	Unknown	None	Asymptomatic	Yes (810)	Serology +(1.32)	None	Lost to follow-up
13	M, 89	June 2018	Farmer	Hypereosinophilic syndrome	Skin	Yes (1700)	Serology +(14.35)	None	Dead
14	M, 80	January 2019	Unknown	None	Asymptomatic	Yes (1660)	Serology +(10.10)	Albendazole	Seroreversion
15	M, 61	January 2019	Printer	Gastroesophageal reflux	Digestive	Yes (510)	Serology +(2.90)	Ivermectin	Seroreversion

COPD: Chronic obstructive pulmonary disease.

**Table 3 pathogens-09-00601-t003:** Clinicoepidemiological characteristics of imported strongyloidiasis cases at Vega Baja Hospital.

Case	Sex, Age in Years	Date of Diagnosis	Country of Case Origin	Occupation	Comorbidities and Risk Factors	Clinical Manifestations	Eosinophilia (Eosinophils/mm^3^)	Method of Diagnosis (in Serology, Optical Density Index)	Strongyloidiasis Treatment	Outcome
1	M, 70	May 1999	Algeria	Unknown	Bladder cancer, asthma Corticosteroids	Digestive	Yes (670)	Larvae in duodenal biopsy; serology not requested	Thiabendazole	Lost to follow-up
2	F, 80	August 2011	Dominican Republic	Housekeeper	None	Digestive	Yes (3000)	Larvae in fresh stool samples and duodenal fluid; serology result unknown	Ivermectin	Clinical recovery
3	F, 78	July 2015	UK	Salesperson	COPD	Digestive	No	Serology +(1.73)	Ivermectin	Lost to follow-up
4	M, 57	September 2015	Ecuador	Metallurgy	Biliary disease	Digestive	Yes (1000)	Serology +(3.33)	None	Lost to follow-up
5	M, 38	November 2015	Bolivia	Farmer	Chagas	Asymptomatic	No (140)	Serology +(1.67)	Ivermectin	Lost to follow-up
6	M, 51	December 2015	Bolivia	Construction worker	Chagas	Asymptomatic	Yes (1300)	Serology +(13.33)	Ivermectin	Serological response
7	M, 52	January 2016	Bolivia	Unknown	Chagas	Asymptomatic	Yes (1390)	Serology +(13.73)	Ivermectin	Serological response
8	M, 52	March 2016	Bolivia	Farmer	Chagas	Asymptomatic	Yes (880)	Larvae in duodenal biopsy; Serology +(15.86)	Ivermectin	Serological response
9	M, 48	May 2016	Bolivia	Construction worker	Chagas	Digestive	Yes (830)	Serology +(7.54)	Ivermectin	Seroreversion
10	F, 50	December 2016	Bolivia	Caregiver	Chagas	Digestive	No (320)	Serology +(7.92)	Ivermectin	Seroreversion
11	F, 35	June 2016	Bolivia	Unknown	None	Asymptomatic	Unknown	Serology +(14.16)	Ivermectin	Serological response
12	F, 50	July 2016	Bolivia	Unknown	Chagas	Asymptomatic	Yes (750)	Serology +(6.71)	Ivermectin	Seroreversion
13	F, 80	October 2016	Jamaica	Nurse	Unknown	Digestive	Yes (1100)	Serology +(2.37)	Ivermectin	Seroreversion
14	F, 37	March 2017	Bolivia	Farmer	None	Digestive	Yes (1000)	Serology +(5.19)	Ivermectin	Re-treatment (ivermectin)
15	M, 68	July 2017	Colombia	Unknown	None	Asymptomatic	Yes (1300)	Serology +(1.85)	Ivermectin	Seroreversion
16	F, 44	February 2018	Morocco	Unknown	Meningioma	Asymptomatic	Unknown	Serology +(1.30)	Ivermectin	Lost to follow-up
17	F, 58	June 2018	Bolivia	Housekeeper	Chagas	Digestive	Yes (700)	Serology +(4.82)	Ivermectin	Seroreversion
18	F, 40	September 2018	Bolivia	Unknown	Chagas	Asymptomatic	Unknown	Serology +(1.15)	Ivermectin	Lost to follow-up
19	F, 57	October 2018	Bolivia	Housekeeper	Chagas	Asymptomatic	Yes (990)	Serology +(6.83)	Ivermectin	Seroreversion
20	F, 37	December 2018	Morocco	Unknown	Hydatidosis	Respiratory	Yes (3960)	Serology +(2.66)	Ivermectin	Seroreversion
21	M, 37	January 2019	Morocco	Shopkeeper	None	Asymptomatic	Yes (510)	Serology +(2.36)	Ivermectin	Lost to follow-up

COPD: chronic obstructive pulmonary disease.

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
