# Peer review of "Strongyloidiasis in Southern Alicante (Spain): Comparative Retrospective Study of Autochthonous and Imported Cases"

_pathogens, 2020, doi:10.3390/pathogens9080601_

Round 1

Reviewer 1 Report

This retrospective study on strongyloidiasis in Spain is informative and interesting. I would like to recommend this article to be published, however, several minor points have to be clarified before publication.

#1 Introduction (lines 64-65): “Autochthonous cases also occur in Spain, especially in the Mediterranean basin (Valencian region and Murcia). The vast majority are restricted to a specific area of La Safor and Marina Alta 65 regions”

Why do autochthonous strongyloidiasis cases occur in these specific areas in Spain? Are people living in this particular regions are different from others? Readers outside Spain (including this reviewer) would not have an idea why this epidemiology.

#2 Results (lines 104-105): “Twenty-nine patients were serologically tested for HIV, all with negative results.”

On the background of the patients, HIV infection was mentioned but HTLV-1 was not. Didn’t the authors test patients for HTLV-1? Latin American countries are endemic for the virus, e.g., AIDS Res Hum Retroviruses. 2014 Sep;30(9):856-62. doi: 10.1089/AID.2013.0128., and it is well known that HTLV-1 exacerbates strongyloidiasis.

Reviewer 2 Report

The authors report the results of a study carried out in a Southern European setting, addressing a neglected topic, such as stongyloidiasis. The study was focussed on autochthonous and imported cases, analyzing clinical, epidemiological and microbiological characteristics of all patients.

The number of observed cases at Vega Baja Hospital, a regional Spanish hospital (Southern Alicante), from 1999 to 2019 is limited (n= 36, of whom 21 were imported cases). The provided data (coexistence of autochthonous and imported cases of strongyloidiasis) are of some interest although not unexpected.

It would be interesting to have information on the human T cell lymphotropic virus (HTLV) serostatus of the included subjects.

Limitations of the study are discussed by the authors.

Some observations:

Results

Line 100. The authors report that three subjects were treated with steroids and had hyperinfection syndrome. Was the steroid treatment prescribed for comorbidities or for symptoms (e.g. pruritus) related to strongyloidiasis? In Table 2, there are only 2 cases of hyperinfection syndrome. Was the third of the three mentioned cases a disseminated strongyloidiasis (case n.8)?

Materials and Methods

Line 255 I suggest to write as follows:  The serological study was carried out at the Spanish National Centre for Microbiology, Instituto Nacional de Salud Carlos III Majadahonda, Madrid

Reference n. 20: Authors names are written in block letters. Please correct.

Reference list I could find was limited to n. 23 !
